# Tiny-CAD-Coder: Leveraging Pre-trained Code Models for Improved CAD Generation

## Abstract

Generative methods for Computer-Aided Design (CAD) are an emerging challenge with broad implications in engineering and manufacturing. Prior work has used direct tokenization or JSON representations and fine-tuned large language models (LLMs), but these approaches are limited to the text and image modalities of the LLM, rely on expensive feedback loops, and struggle with validity. We argue that CAD is inherently code-like: it consists of ordered command sequences with variable parameters, closely resembling programming languages. Building on this insight, we introduce Tiny-CAD-Coder, a framework that fine-tunes pre-trained code models for CAD generation and adapts to diverse input modalities through prefix tuning. By representing construction histories as Python code, our method exploits the syntactic and semantic priors of code models, while prefix embeddings provide a lightweight and extensible interface to condition on B-Rep, text, images, or other structured inputs that are not covered by LLMs used in prior works. Our experiments show that a 1B parameter code model matches or outperforms fine-tuned 7B-parameters LLMs and multi-shot prompting with 450B models. We also contribute a dataset of 180k CAD code samples derived from Omni-CAD. Our results establish code models with prefix tuning as an efficient and general foundation for multi-modal CAD generation tasks.

## 1 Introduction

Generative AI is increasingly impacting the engineering design domain with CAD as its central tool (Hirz et al., 2011; Sharma et al., 2023; Berger et al., 2025b), yet the creation of parametric models requires manual labor and expertise-intensive (Regassa Hunde & Debebe Woldeyohannes, 2022). This has motivated research into learning-based CAD automation (Berger et al., 2025b). Automating CAD requires generating editable, parametric sequence rather than meshes or voxels, as engineering applications demand modifiability and reusability (Camba et al., 2016; Willis et al., 2021).

Recent work represents CAD as JSON and adapts general-purpose LLMs (Wang et al., 2025; Xu et al., 2025), but overlooks the inherently structured, code-like nature of CAD construction sequences. Other publications have explored using a Python CAD library to generate CAD objects iteratively with commercial LLMs such as GPT-4 (Alrashedy et al., 2025; Mallis et al., 2025; Ocker et al., 2025) or fine-tuning open-source Vision-Language-Models (VLM) (Doris et al., 2025; He et al., 2025). These approaches leverage existing models and their multi-modal capabilities and are therefore easily accessible. But they do not generalize beyond text and image conditions, report issues in generating valid geometries, and incur high API costs.

Our approach differs from these works by (1) using a pre-trained code model as the foundation for CAD generation instead of a pre-aligned VLM. We argue that models pre-trained on large amounts of code are better suited for CAD generation than general-purpose LLMs, as they inherently understand the structured nature of code-like data. (2) Unlike prior works that restrict evaluation to modalities already well supported by off-the-shelf LLMs our method uses prefix tuning (Li & Liang, 2021) to easily adapt our model to new input modalities instead of relying on fine-tuning a pre-aligned model. And (3) we remove computationally expensive iterative feedback loops, enabling efficient one-shot CAD generation.

To this end, we introduce Tiny-CAD-Coder, which leverages DeepSeek-Coder-1B (Guo et al., 2024) for efficient CAD generation. First, we pretrain the model on CAD code using the self-supervised next-token prediction objective, which allows the model to learn the structure and semantics of CAD construction sequences. Then, we train an alignment model on various modalities such as B-Rep embeddings. Finally, using prefix tuning, we condition our model on various input modalities beyond generic text and image inputs. Our experiments demonstrate that a 3B parameter code model surpasses the performance of fine-tuned 7B LLMs and multi-shot 450B models.

In short, our contributions are as follows:

- We establish that CAD is best represented as code, and validate this by comparing different data representations on sequence lengths and ablating LLMs and code models for CAD generation.
- We introduce Tiny-CAD-Coder, a parameter-efficient approach that leverages DeepSeek-Coder with prefix tuning. Our 1B parameter model matches or surpasses the performance of larger models. Further, it surpasses the performance of previous GAN-based CAD generation methods.
- We demonstrate that our approach enables the use of various input modalities, that have not been explored in previous works, showcasing its versatility across CAD generation tasks.

## 2 RELATED WORK

### 2.1 INPUT MODALITIES

Recent publications tackle Text-to-CAD or Image-to-CAD generation tasks, where the input is a text description or an image of the desired CAD object, because existing VLMs already support these data modes. However Berger et al. (2025b) points out that engineers actually prefer different CAD tasks to be performed by AI such as reverse engineering B-Rep objects or generating variants from a given CAD object, rather than generating CAD from text or images.

Boundary Representation (B-Rep) is a key data format commonly used as a vendor-neutral file format, facilitating interoperability between different CAD systems. This interoperability comes with a loss of information, because B-Rep only contains the faces and edges of a 3D object, but the construction history is lost. Reverse engineering the construction history given only the outer shell a significant challenge (Dupont et al., 2022).

Further, this task is more challenging than Text-to-CAD or Image-to-CAD generation, as there is no pretrained LLM aligned with B-Rep embeddings available, whereas LLMs and VLMs come already aligned with text and image embeddings.

### 2.2 GAN-BASED CAD GENERATION

Early approaches (Para et al., 2021; Wu et al., 2021; Xu et al., 2023; 2022) for generating CAD construction sequences used Latent-GANs, where a GAN (Goodfellow et al., 2014) learns a latent representation of CAD objects that is decoded into sequences by a Transformer model. These models are limited to unconditional generation and cannot incorporate external inputs such as text or B-Rep data, which are essential for tasks like Text-to-CAD and B-Rep-to-CAD generation.

### 2.3 LLM-BASED CAD GENERATION

Recent approaches have discovered LLMs for CAD generation. There are two main criteria to classify the approaches: training method is one of the following: no training, fine-tuning and full training. Second is the choice of representation for CAD data. This is one of the following: JSON representations of CAD objects, Python code representations based on a library such as CadQuery (Urbanczyk et al., 2024), and a CAD-specific tokenized representation (Wu et al., 2021). A structured overview of recent literature is given in Table 1.

**No Training** The multi-task capabilities of commercial LLMs such as GPT-4 allow for CAD generation without additional training (Alrashedy et al., 2025; Mallis et al., 2025; Ocker et al., 2025)

Table 1: Comparison of CAD generation approaches using LLMs grouped by training method and data representation.

| Training Method | Data Representation | Publications |
|---|---|---|
| Zero-Shot | Code | Alrashedy et al. (2025); Mallis et al. (2025); Ocker et al. (2025) |
| Fine-Tune | JSON | Xu et al. (2025); Wang et al. (2025) |
| Fine-Tune | Code | Doris et al. (2025); He et al. (2025) |
| Full Training | Tokens | Khan et al. (2024); Alam & Ahmed (2025) |

and are therefore quick to implement. However, the LLMs are not specifically trained for CAD tasks and hence tend to output invalid CAD and therefore require iterative feedback loops. For example Mallis et al. (2025) report a almost no correct CAD objects in one-shot inference. While these approaches show promise, they are computationally expensive and slow due to the iterative nature of the feedback loop and the large size of the models.

**Fine-Tuning**  Fine-tuning approaches typically use 7B LLMs or VLMs to generate Python or JSON representations of CAD objects, with recent work exploring VLM fine-tuning for CAD tasks using CadQuery (Urbanczyk et al., 2024) or JSON-based formats (refer to Table 1). Doris et al. (2025) makes use of a VLM to encode an input image and then generates the corresponding CAD construction sequence. However, these approaches still rely on large LLMs or VLMs, which are computationally expensive and inefficient for the narrow scope and structured nature of CAD generation tasks and limit conversion tasks to the modality of the base models.

**Full Training**  In contrast to fine-tuning, full training involves training a Transformer model for CAD generation from scratch. This approach has been explored in Khan et al. (2024); Alam & Ahmed (2025), where the model is trained on a large corpus of CAD data to learn the underlying structure and semantics of CAD construction sequences. This is feasible because the tokenized representation of CAD data is compact, allowing for a small context window of 512 tokens and less than 100M trainable parameters.

### 2.4 Other CAD Generation Methods

Other approaches directly generate B-Rep (Boundary Representation) (Jayaraman et al., 2021; Zhang et al., 2024) or 3D meshes (Nash et al., 2020) data without relying on construction sequences, but these 3D object representations lack the editability required for engineering design with CAD (Vukašinović & Duhovnik, 2019). Hence, we do not consider these approaches in our work.

## 3 Data

**Data format.** We only consider CAD data containing construction histories as such parametric data is best suitable for engineering design. In contrast to prior JSON-based approaches, we represent CAD data as Python code to capture its hierarchical, sequential, and code-like nature. Accordingly, our ablation study in Section 7.1 confirms that code models outperform general-purpose LLMs for CAD generation.

Furthermore, compared to JSON representations used in prior work, our Python code representation uses on average 73.6% less tokens per sample, reducing sequence length and hence computational requirements (see Section A.1 of the appendix)

**Preprocessing pipeline.** We normalize each model to a unit cube and center it at the origin to remove scale and translation bias. We round all floating-point values to two decimal places to reduce

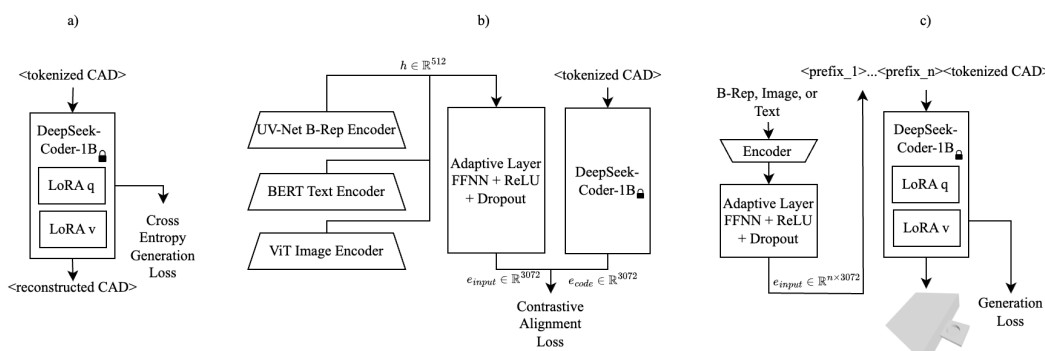

Figure 1: Overview of the proposed training procedure. a) Self-supervised pre-training of the base code model on CAD construction sequences to adapt it to CAD-specific syntax and semantics. b) Alignment of modality-specific encoders (B-Rep, text, image) with the code model using a contrastive loss on paired data. c) End-to-end prefix tuning with multimodal inputs to learn task-specific prefixes for conditioned CAD generation while preserving the knowledge from pre-training. The lock icon indicates frozen model parameters.

the input length and apply de-duplication. All geometric processing is done using the RapidCAD-Py[1] library.

Then we use a custom script to reverse-engineer Python code that generates the CAD objects and generate the images and B-Rep objects for the Image-to-CAD and B-Rep-to-CAD tasks. The text descriptions are taken from Xu et al. (2025).

Finally, we partition the dataset into 90% training, 5% validation, and 5% testing.

## 4 METHODOLOGY

We use DeepSeek-Coder-1B (Guo et al., 2024) as the base code generation model. It is pre-trained on diverse programming languages, including Python, providing a strong understanding of structured and hierarchical code patterns. To efficiently adapt the model under computational constraints, we apply LoRA (Hu et al., 2021) with rank $r = 32$ and scaling factor $\alpha = 64$, modifying only the query and value projection matrices.

**Self-supervised Pre-training** The objective of the pre-training stage (Figure 1a) is to adapt the base DeepSeek-Coder-1B model to CAD-specific code patterns by continuing training on a large corpus of CAD construction sequences. The model is trained with the standard autoregressive language modeling loss:

$$\mathcal{L}_{\text{pretrain}} = -\sum_{t=1}^{T} \log P(x_t \mid x_{<t}; \theta),$$

where $x_t$ denotes the token at time step $t$ and $\theta$ are the model parameters. This continued pre-training enables the model to capture CAD-specific syntax, geometric primitive definitions and construction sequence patterns. Then the model has to learn this once and the task-specific prefixes can be learned efficiently in the next steps. We will re-use the decoder for different tasks, shortening the overall training time.

**Alignment Training** The goal of alignment training (part b) in Figure 1 is to integrate geometric information from the encoder with the DeepSeek-Coder-1B embedding space, facilitating effective cross-modal conditioning. To achieve this, we learn projection layers that map features into prefix tokens compatible with DeepSeek-Coder-1B We employ a contrastive alignment loss of the form:

$$\mathcal{L}_{\text{align}} = -\log \frac{\exp(\text{sim}(z_g, z_c)/\tau)}{\sum_j \exp(\text{sim}(z_g, z_j)/\tau)},$$

---

[1]https://www.github.com/anonymized_during_review

where $z_g$ is the geometric embedding, $z_c$ is the corresponding code embedding, $\tau$ is a temperature parameter, and *sim* is the cosine similarity function. We use a decay in the alignment loss weight so during the beginning of the training, aligning the embeddings is prioritized. Our experiments showed that this helps in stabilizing training and leads to better convergence.

**End-to-End Prefix Tuning**    The final training stage in Figure 1c involves end-to-end prefix tuning, where the objective is to learn task-specific prefixes for conditioned CAD generation while preserving the knowledge from pre-training. To enable conditioning on inputs, we use task-specific encoders to learn a latent embedding from the input, which is then projected by a lightweight adaptive layer into a sequence of prefix embedding $P \in \mathbb{R}^{k \times d}$, where $k$ is the prefix length and $d$ the hidden dimension. The prefix embedding is split into prefix tokens and is prepended to the code token embeddings, enabling cross-modal conditioning without modifying the decoder weights:

$$h_0 = [P; \text{Encoder}(x_{\text{input}}); \text{Embed}(x_{\text{code}})].$$

We optimize a combined generation and reconstruction loss:

$$\mathcal{L}_{\text{total}} = \mathcal{L}_{\text{gen}} + \lambda \mathcal{L}_{\text{recon}},$$

where

$$\mathcal{L}_{\text{gen}} = -\sum_{t=1}^{T} \log P(y_t \mid y_{<t}, x_{\text{input}}; \theta, P),$$

and

$$\mathcal{L}_{\text{recon}} = \|\text{UV-Net}(\text{Execute}(y)) - \text{UV-Net}(x_{\text{input}})\|_2^2.$$

The generation loss encourages accurate conditional CAD code generation, while the reconstruction loss ensures that the generated model's geometry remains consistent with the input.

## 4.1 Implementation Details

Our experiments show that a range of 8 to 16 prefix tokens is sufficient to capture the necessary context for CAD generation, balancing expressiveness and efficiency. Increasing the number of tokens beyond this range did not improve the loss, while fewer tokens led to performance degradation. Training was performed on four Nvidia L40S GPUs with a combined 196 GB of GPU memory and took approximately 12 hours to complete. The CAD code required a sequence length of 1024 tokens to cover 86% of the training data, which we found sufficient to capture the complexity of CAD construction sequences. For details on hyperparameters refer to Section 7.2.

## 5 Experiments

For our experiments, we use our Tiny-CAD-Coder model that has been pretrained on CAD data. For all experiments, we use the same pretrained decoder, demonstrating its ability to adapt to different tasks. In our experiments, we first evaluate our decoder's capability to generate CAD objects from scratch. We then demonstrate its adaptability by fine-tuning the model to generate CAD objects from B-Rep representations and text descriptions.

## 5.1 Evaluation Metrics

To evaluate our model quantitatively, we use geometric and syntactic metrics. The geometric metrics assess the quality of the generated CAD objects, while the syntactic metrics evaluate the correctness of the generated construction sequences.

The **Chamfer Distance (CD)** (Fan et al., 2016) between two shapes $S_1$ and $S_2$ finds for each point $(x, y)$ in a point cloud its nearest neighbor in the opposite point cloud and sums the squared distances:

$$\text{CD}(S_1, S_2) = \frac{1}{|S_1|} \sum_{x \in S_1} \min_{y \in S_2} \|x - y\|_2^2 + \frac{1}{|S_2|} \sum_{y \in S_2} \min_{x \in S_1} \|x - y\|_2^2. \tag{1}$$

The **Normal Consistency (NC)** metric evaluates the geometric consistency of generated CAD models by comparing surface normal vectors. For each face in the generated geometry, we compute the angular difference between its normal vector and the corresponding face in the ground truth model:

$$\text{NC} = \frac{1}{N} \sum_{i=1}^{N} \cos^{-1}(\mathbf{n}_i^{\text{pred}} \cdot \mathbf{n}_i^{\text{gt}}), \tag{2}$$

where $\mathbf{n}_i^{\text{pred}}$ and $\mathbf{n}_i^{\text{gt}}$ are the normalized normal vectors of the $i$-th face in the predicted and ground truth models, respectively.

The **Invalid Ratio (IR)** is the ratio of invalid sequences to the total number of generated sequences. A sequence is considered invalid if it cannot be processed by a CAD system due to geometric or syntactic errors.

The **Intersection over Union (IoU)** metric measures the overlap between the reconstructed model $\hat{M}$ and the ground truth model $M$. It is defined as

$$\text{IoU}(\hat{M}, M) = \frac{\left| \hat{M} \cap M \right|}{\left| \hat{M} \cup M \right|}, \tag{3}$$

where $|\cdot|$ denotes the volume of the respective region. The IoU value ranges from 0 (no overlap) to 1 (perfect overlap).

The **CodeBLEU Score** (Ren et al., 2020) is a specialized metric for evaluating code generation that extends traditional BLEU by incorporating syntax tree matching and data flow analysis. Unlike standard BLEU, which only considers n-gram overlap, CodeBLEU captures the structural and semantic properties of code:

$$\text{CodeBLEU} = \alpha \cdot \text{BLEU} + \beta \cdot \text{BLEU}_{\text{weight}} + \gamma \cdot \text{Match}_{\text{ast}} + \delta \cdot \text{Match}_{\text{df}}, \tag{4}$$

where $\text{Match}_{\text{ast}}$ measures abstract syntax tree similarity and $\text{Match}_{\text{df}}$ evaluates data flow consistency. We report the CodeBLEU as an average of both valid and invalid sequences, as it reflects the model's overall ability to generate syntactically correct and semantically meaningful CAD code.

## 5.2 SEQUENCE COMPLETION

Sequence completion involves predicting the next elements in a partially observed sequence. In the context of CAD generation, this means inferring the missing CAD commands from a given partial sequence. This tasks forms the base for the other tasks, as the model has to learn the structure and semantics of CAD commands and their parameters.

## 5.3 B-REP-TO-CAD GENERATION

This follows the setup in Figure 1c. The encoder is based on the UV-Net architecture (Jayaraman et al., 2021) and embeds B-Rep objects into a latent vector $z$ by converting B-Reps into a face-adjacency graph and processing them with graph neural networks. We add batch normalization between convolutional layers because we experienced numerical instability during our experiments. The latent vector $z$ is passed into our decoder model as prefix tokens. Additionally to the memory, the decoder receives context tokens $C_{1:i}$ from the original construction sequence as context to infer the following tokens. For our experiments we set $i = 5$.

Given the B-Rep input $B$ and the pre-trained parameters $\theta$, the model is trained with cross-entropy loss between the original sequence of CAD commands $C$ and the predicted sequence $\hat{C}$:

$$\mathcal{L}_{\text{B-Rep}} = -\sum_{t=1}^{T} \log P(c_t \mid c_{<t}, B; \theta), \tag{5}$$

where $c_t$ represents the CAD command token at time step $t$. Both models were trained until convergence, defined as the point at which the improvement in validation loss was less than $\Delta = 0.001$ for $n_{\text{patience}} = 5$ consecutive epochs. We expect the pretrained decoder to accelerate the task-specific training process, as the model has already learned the underlying structure of CAD commands and their parameters during pre-training.

Table 2: Unified results across tasks. Dashes indicate metrics not reported in the original publications. We evaluate on the three established CAD generation tasks: Sequence Completion, Text-to-CAD, and Image-to-CAD. CadCodeVerify* (Alrashedy et al., 2025) takes as input text plus image data and uses iterative refinement. Our model achieves competitive or superior performance across all tasks while using significantly fewer parameters than prior works.

| Model | CodeBLEU ↑ | CD ↓ | NC ↑ | IR ↓ | IOU ↑ |
|---|---|---|---|---|---|
| **Sequence Completion** | | | | | |
| Ours | 0.85 | **1.00** | 96.0 | **3%** | 51.4 |
| Berger et al. (2025a) | – | 1.01 | – | 8% | – |
| Mamba-CAD (Li et al., 2025) | – | 1.03 | – | 8% | – |
| DeepCAD (Wu et al., 2021) | – | 69.7 | – | 40.2% | – |
| TM-CAD (Li et al., 2025) | – | 82.3 | – | 44.3% | – |
| **Text-to-CAD** | | | | | |
| Ours | 0.75 | **2.22** | 99.0 | **0.25%** | – |
| Berger et al. (2025a) | – | 2.58 | – | 4.4% | – |
| Text2CAD (Khan et al., 2024) | – | 2.65 | – | 0.93% | – |
| CAD-GPT (Wang et al., 2025) | – | 2.83 | – | 7.43% | – |
| **Image-to-CAD** | | | | | |
| Ours | 0.79 | **5.39** | 97.2 | 14.5% | 58.8 |
| CadCodeVerify* (Alrashedy et al., 2025) | – | – | – | 6.0% | **94.1** |
| CadCoder (Doris et al., 2025) | – | – | – | **0.0%** | 67.5 |
| CAD-GPT (Wang et al., 2025) | – | 9.77 | – | 1.61% | – |
| **B-Rep-to-CAD** | | | | | |
| Ours | 0.62 | **2.05** | 95.0 | **6.99%** | 49.9 |
| Berger et al. (2025a) | – | 2.64 | – | 10.1% | – |
| BRep2Seq (Zhang et al., 2024) | – | 3.35 | – | 11.8% | **70.3** |

## 5.4 TEXT-TO-CAD GENERATION

Again, we follow the setup in Figure 1c with a text encoder. We use the pre-trained BERT text encoder to embed text descriptions into a latent vector $z$. The text description data is taken from Xu et al. (2025) and our Python CAD code to train first the alignment and then the prefix tuning stage. We re-use the same pre-trained decoder model as in the previous task, reducing the overall training time.

## 5.5 IMAGE-TO-CAD GENERATION

We follow the setup in Figure 1c with an image encoder. We use a pre-trained Vision Transformer (ViT) (Wu et al., 2020) to embed images into a latent vector $z$ and pass this latent vector as prefix tokens to our decoder. We use the images provided by Xu et al. (2025) and our Python CAD code to train first the alignment and then the prefix tuning stage. We re-use the same pre-trained decoder model as in the previous tasks, reducing the overall training time.

## 6 RESULTS

We evaluate Tiny-CAD-Coder across four CAD generation tasks: Sequence Completion, B-Rep-to-CAD, Text-to-CAD, and Image-to-CAD. Our results are summarized in Table 2. Qualitative results for the Text-to-CAD and B-Rep-to-CAD tasks are shown in Figures 2 and 3, respectively.

**CodeBLEU** Among code-based CAD generation methods, we are the first to report CodeBLEU. Scores between 0.62 and 0.85 indicate that our model generates syntactically correct and semantically meaningful CAD programs, validating the choice of a code-centric representation.

**Invalid Ratio** On the widely reported invalid ratio, our model consistently outperforms prior work across all tasks except Image-to-CAD. In that setting, CadCodeVerify (Doris et al., 2025) achieves

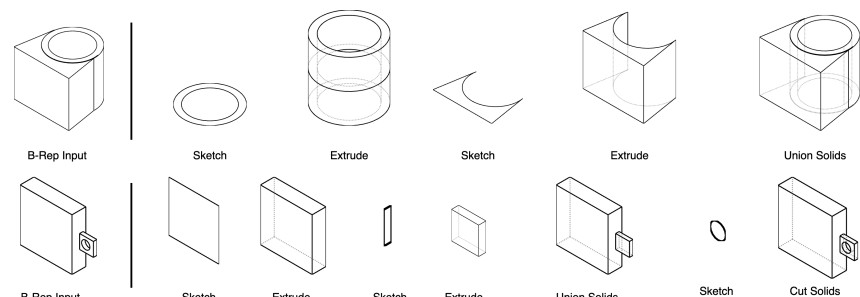

Figure 2: Qualitative results of the B-Rep-to-CAD generation task. The model takes a B-Rep representation without construction history (left) as input and generates a corresponding CAD construction sequence (right).

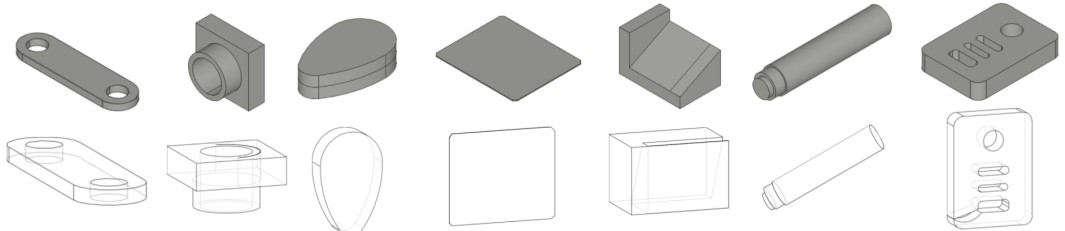

Figure 3: Randomly sampled qualitative results of the Image-to-CAD generation task. The model uses a visual transformer architecture to generate CAD code from an image input. Top row contains the input image and bottom row the rendered CAD model. Columns five and seven show error cases.

a perfect score of 0.0 due to its specialized verification pipeline. Nevertheless, our approach yields substantially lower invalid ratios in Text-to-CAD and B-Rep-to-CAD, demonstrating improved robustness without post-hoc correction.

**Geometric Quality** For Chamfer Distance and Normal Consistency, our approach matches or surpasses larger fine-tuned VLMs while using significantly fewer parameters. Notably, our 1B parameter model achieves comparable or better results than fine-tuned 7B LLMs and even multi-shot prompting with 450B models.

**Efficiency** Our framework achieves these results without iterative feedback loops, reducing inference latency and compute cost. This establishes code models with prefix tuning as a parameter-efficient and scalable foundation for CAD generation.

**Qualitative Evaluation** Figures 2–4 provide visual comparisons. Generated CAD objects closely match ground truth sequences in geometry and topology, with failure cases primarily arising from minor syntactic errors (Figure 6).

# 7 ABLATION STUDY

## 7.1 CHOICE OF BASE MODEL

To validate our hypothesis that code models are better suited for CAD generation than general-purpose LLMs, we conduct an ablation study comparing models with 3B parameters or less, because for narrow tasks small, specialized models achieve comparable performance to large models and are more economical to deploy (Belcak et al., 2025). We pretrain all models on the same CAD dataset on the sequence completion task on 1% of the data, evaluating performance metrics, maximum feasible batch size (Max BS) and speed in optimizer steps per second, ceteris paribus. As a control, we also evaluate DeepSeek-Coder-1B with random weights initialization. The results in Table 3 show that code models perform better than general-purpose LLMs, with DeepSeek-Coder-1B achieving the best trade-off between performance and efficiency. Pre-training on code yields a clear performance

gain over random initialization, demonstrating that the improvement stems from the prior knowledge encoded in the model.

Table 3: Comparison of LLMs and code models for CAD generation. Batch size refers to the maximum feasible batch size before out of memory errors occur. Speed is measured in optimizer steps per second. Our chosen base model, DeepSeek-Coder-1B, achieves the best trade-off.

| Model Name | CodeBLEU | CD | IR | Max BS | Speed |
|---|---|---|---|---|---|
| StarCoder-3B (Lozhkov et al., 2024) | 0.69 | 4.03 | 64.6 | 10 | 1.20 |
| CodeGemma-2B (Team et al., 2024) | 0.63 | 22.0 | 78.2 | 4 | 0.3 |
| DeepSeek-Coder-1B (Guo et al., 2024) | **0.78** | 10.8 | **39.2** | **16** | 1.81 |
| QwenCoder2.5-3B (Hui et al., 2024) | 0.68 | **1.1** | 71.9 | 2 | 0.2 |
| Gemma2-3B (Riviere et al., 2024) | 0.67 | 28.4 | 73.8 | 5 | 0.82 |
| StableLM-3B (Tow et al., 2024) | 0.73 | 15.8 | 47.8 | 8 | – |
| DeepSeek-Coder-1B (rand. init.) (Guo et al., 2024) | 0.66 | 20.0 | 70.6 | 16 | 1.81 |

## 7.2 HYPERPARAMETERS

We used gridsearch to optimize the hyperparameters on a small proxy models (Liu et al., 2024). The following hyperparameters were optimized: learning rate, batch size, prefix length. The final configuration used a learning rate of $1 \times 10^{-4}$, batch size of 8, and 16 prefix tokens. We used the AdamW optimizer with an initial learning rate of 0.001, weight decay of 0.01, and a cosine learning rate schedule. The final hyperparameters can be found in the supplementary source code.

## 8 CHALLENGES AND LIMITATIONS

Despite the promising results of Tiny-CAD-Coder, several challenges and limitations remain that affect the practical deployment of code-based CAD generation systems.

**Invalid code.** Unlike other publications like Doris et al. (2025), we found generating syntactically valid code to be a greater challenge. Our model tends to make small errors such as missing parentheses or confusing commas for periods, which can render the entire sequence invalid. For examples and a more detailed analysis, please refer to Section B of the appendix.

**Bias toward simple geometries.** The model tends to generate simpler CAD objects, likely due to their overrepresentation in the training data and lower risk of invalidity. Valid sequences average 97.4 characters, invalid ones 131.1 (Mann–Whitney (Mann & Whitney, 1947) $U$ test, $p = 5.5 \times 10^{-4}$). Similar trends were observed in prior work (Wu et al., 2021).

**Broken geometry.** Even when a generated construction sequence is syntactically correct, the resulting geometry may be topologically invalid. These issues arise when the model learns the statistical distribution of construction sequences, resulting in outputs that are syntactically valid but geometrically flawed.

## 9 CONCLUSION

We presented Tiny-CAD-Coder, a framework that adapts pre-trained code models for parametric CAD generation. By casting construction histories as code and applying prefix tuning, our 1B model matches or exceeds the performance of fine-tuned 7B LLMs and even multi-shot 450B models, while enabling conditioning on diverse modalities that can be expanded in the future. This establishes code models as an efficient and scalable foundation for CAD generation.

Remaining challenges include ensuring syntactic validity, overcoming a bias toward simple geometries, and enforcing geometric consistency. Future work will integrate constraint-aware validation, adopt curriculum strategies to increase design complexity, and extend prefix tuning to modalities such as sketches or design intent. More broadly, our results suggest that domain-specific code models can serve as compact yet powerful backbones for structured generation tasks beyond CAD.

## REPRODUCIBILITY STATEMENT

We provide complete source code and instructions as supplementary material. The package includes dataset loaders for B-Rep, images, and text (`data_loader/`), model definitions and configuration files (`models/`), and evaluation utilities (`modules/`). A `train.py` entry point supports training and inference across all modalities. YAML configuration files store all hyperparameters such as learning rate, batch size, and dataset paths. The included `README.md` details setup steps, dependency installation, and data preparation, including scripts for generating Python code, B-Reps, and images. All reported metrics (CodeBLEU, Chamfer Distance, Normal Consistency, Invalid Ratio, IoU) are computed and logged automatically during evaluation.

Results in the paper can be reproduced by running the provided training configurations using `train.py` with one of the keywords (`--cadcoderseq`, `--cadcoderbrep`, `--cadcodertext`, `--cadcoderimage`). In the YAML configuration, the model training can be set to start training, continue from a checkpoint or run evaluation mode. For more details, please refer to the `README.md` and `config.py`. Training was performed on 4 NVIDIA L40S GPUs (196 GB total memory), but the code also runs on smaller hardware by reducing batch size.

While we provide complete scripts and instructions, some modifications may be required depending on the operating system, hardware setup, or library versions. Researchers may report issues through the repository's issue tracker; we intend to address common problems where feasible.

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

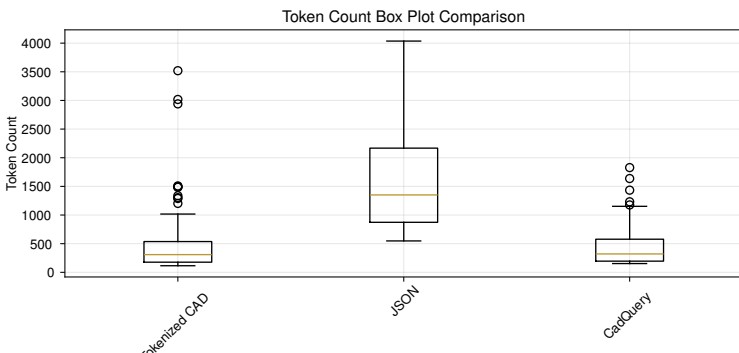

Figure 4: Ablation study comparing CAD representations. Python code is both compact and expressive, enabling the use of pre-trained code models.

# A  ABLATIONS

## A.1  COMPARISON OF CAD REPRESENTATIONS

We next compare alternative ways of representing CAD objects for sequence modeling. Three formats are evaluated: (1) JSON-based representations as in Wang et al. (2025); Xu et al. (2025), (2) tokenized CAD-specific command vocabularies (Wu et al., 2021; Khan et al., 2024; Alam & Ahmed, 2025), and (3) Python code using the CadQuery library.

Figure 4 shows that JSON is the most verbose due to heavy use of punctuation and structural tokens. Tokenized CAD commands are compact, but lack flexibility and human readability. Our Python code representation achieves similar compactness while preserving full expressiveness and compatibility with off-the-shelf code models. This leads to shorter sequences, lower memory requirements, and faster training.

For this comparison we randomly sampled 1000 CAD objects from the DeepCAD dataset (Wu et al., 2021).

## A.2  PREFIX TOKEN USAGE

To verify that the model attends to the prefix tokens rather than relying solely on teacher forcing during training, we logged attention weights between prefix and code tokens throughout training. In additional experiments, we masked the entire code input sequence, forcing the model to rely exclusively on the prefix tokens. Despite this constraint, the model's loss consistently decreased, confirming that the decoder learns to extract useful information from the prefix embeddings. These results validate that the prefix encoder contributes meaningful conditioning signals.

# B  FAILURE CASES

Although Tiny-CAD-Coder reduces invalid outputs compared to prior work, a subset of generated programs still fail due to syntactic or geometric errors. Figure 5 quantifies the main failure categories. Only ~6% of invalid cases stem from geometry inconsistencies, while the majority are simple syntax issues such as missing parentheses, misused punctuation, or undefined variables. Figure 6 shows example excerpts of invalid code.

# LLM USAGE

We used an LLM to assist with grammar correction, typo fixing, and improving the clarity of English text as English is not our first language.

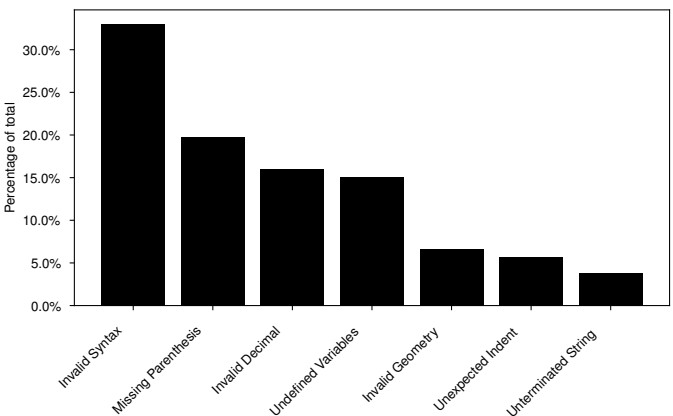

Figure 5: Histogram of invalid code reasons. Most errors are syntactic, indicating that lightweight code-repair methods could substantially reduce invalid ratios.

```
1  [...]
2  wp_sketch0 = app.Workplane(Plane(Vector(-0.73, 0,18, 0), Vector(1, 0, 0),
       Vector(0, 0, 1)))
3  [...]
```

```
1  [...]
2  solid0=wp_sketch0.add(loop0).extrude(0.02)
3  solid=solid0
4   Generating a workplane for sketch 1
5   [...]
```

```
1  [...]
2  loop1=wp_sketch1.moveTo(0.4,, 0).circle(0.37)
3   [...]
```

Figure 6: Example excerpts of syntactically invalid CAD code. (Top) An invalid decimal due to a comma causes an `InvalidDecimal` exception. (Middle) A missing # comment symbol triggers an `IndentationError`. (Bottom) An extra comma causes a `SyntaxError`.

