# OpenReview forum: "Tiny-CAD-Coder: Leveraging Pre-trained Code Models for Improved CAD Generation"
_ICLR.cc/2026/Conference — ICLR 2026 Conference Desk Rejected Submission_

### Official Review · Reviewer_zDLN · 2025-10-20

**Soundness:** 2
**Presentation:** 2
**Contribution:** 2
**Rating:** 2
**Confidence:** 5

**Summary:**

The paper presents Tiny-CAD-Coder - a novel multi-modal CAD generation method. The proposed model inputs texts, or images, or B-Reps and outputs a CAD model in form of Python code. The main idea of introducing new modalities is the prefix tuning of DeepSeek-Coder-1B. The model is claimed to achieve state-of-the-art results in 4 CAD generation tasks.

**Strengths:**

- The model supports 3 input modalities: text, image, and B-Rep.
- Tiny-CAD-Coder is claimed to outperform competitors in the task of CAD completion, and CAD reconstruction.

**Weaknesses:**

- Most of the contributions listed in introduction are far from being novel:
  - L.048: *"(1) using a pre-trained code model as the foundation for CAD generation instead of a pre-aligned VLM"* Actually, a lot of works that are already cited use just general purpose LLMs, with tons of Python code in their training set mixture, like Llama in CAD-MLLM, or Qwen2.5 in CAD-Recode. Even if specialized code LLMs matters, Text-to-CadQuery paper benchmarks CodeGPT, or CADMium involves Qwen2.5-Coder, or CodeLlama in CadCodeVerify paper.
  - L.050: *"(2) Unlike prior works that restrict evaluation to modalities already well supported by off-the-shelf LLMs our method uses prefix tuning"* Actually, several works in LLM-based CAD generation supports additional modalities via prefix tuning. Namely CAD-Recode [ICCV 2025, code available], Cadrille [arxiv, May 2025, code available], CAD-MLLM [arxiv, November 2024, data available] plug point cloud modality to pre-trained LLMs. Most importantly, 2 latest papers even support 3 modalities (image, text, poin cloud) in the same model.
  - L.053: *"(3) we remove computationally expensive iterative feedback loops, enabling efficient one-shot CAD generation."* Actually, all reported (and ignored) baselines already do not use any feedback loops, DeepCAD, Text2CAD, CAD-Coder, CAD-MLLM, CAD-GPT, Cadrille.
  - L.063 *"We establish that CAD is best represented as code"* I think this is already established in recent Text-to-CadQuery or CAD-Recode papers, comparing DeepCAD representation to CadQuery Python code.

- Main results are order of magnitude worse than state-of-the-art.
  - Table 2, text-to-cad. The CD = 2.65 and IR = 0.93 numbers for Text2CAD dataset are used from original Text2CAD paper, assuming that CD is multiplied by 100. The proposed Tiny-CAD-Coder reports CD = 2.22. In this terms Text-to-CadQuery paper reports CD = 1.11, and Cadrille reports CD = 0.39. Other papers with better results may include CAD-Coder [arxiv, 2505.19713] and CADMium [arxiv, 2507.09792].
  - Table 3, image-to-cad. The CD = 9.77 is the value from CAD-GPT paper for DeepCAD dataset. The proposed Tiny-CAD-Coder reports CD = 5.39. In this terms Img2CAD reports CD = 1.60, CADCrafter [CVPR 2025] reports CD = 0.72, and Cadrille reports CD = 0.21.

- Important details about training and test datasets are missing.
  - Correct me if I'm wrong, but regarding datasets I only see L.025: *"We also contribute a dataset of 180k CAD code samples derived from Omni-CAD."*, L.255: *"we use our Tiny-CAD-Coder model that has been pretrained on CAD data"*, L.732: *"For this comparison we randomly sampled 1000 CAD objects from the DeepCAD dataset "*. From these 3 sentences I actually don't understand training and test datasets.
  - Actually, regarding 180k Omni-CAD dataset, I only see information in abstract, and not in main paper. If Omni-CAD dataset from CAD-MLLM paper is used, than the comparison on image-to-cad and text-to-cad with CAD-MLLM method is missing.
  - Looks like in Table 2, some test sets are DeepCAD dataset, but some other are not, e.g. CADCodeVerify uses CADPrompt dataset. In this case it is misleading to put results on multiple datasets to the same table.

**Questions:**

- What are the actual novelties, that are not mentioned in previous works?

- Can Tine-CAD-Coder be compared to CAD-MLLM, Cadrille, Text-to-CadQuery, Img2CAD, and CADCrafter papers?

- What datasets are used for training and benchmarking, DeepCAD or Omni-CAD?

- Why adding CodeBLEU and NC columns to Tab. 2 without calculating them for the open-sourced baseline methods?

- How exactly is the sequence completion benchmark constructed? What is the point of calculating CD on CAD completion? I think, there are infinite amount of correct completions with any CD from 0 to +inf.

---

### Official Review · Reviewer_Uq9J · 2025-10-24

**Soundness:** 1
**Presentation:** 2
**Contribution:** 2
**Rating:** 4
**Confidence:** 4

**Summary:**

This paper proposes a parameter-efficient framework that leverages pre-trained code language models—specifically DeepSeek-Coder-1B—for parametric CAD generation by representing construction histories as executable Python code.

The core technical insight proposed by this paper is that CAD sequences are inherently code-like, and thus benefit more from code-specific pre-training than from general-purpose or vision-language LLMs.

The authors adopt a three-stage training pipeline:

(1) self-supervised pre-training on CAD code to internalize domain syntax and semantics,

(2) contrastive alignment of modality-specific encoders with the code model’s embedding space,

and (3) lightweight prefix tuning to condition generation on diverse inputs without modifying the frozen decoder.

**Strengths:**

1. The main topic and proposed contributions are clearly expressed and easy to follow.

**Weaknesses:**

**Major problems:**

1. The claim in Lines 63-65 is not sound enough:

> "We establish that CAD is best represented as code, and validate this by comparing different data representations on sequence lengths and ablating LLMs and code models for CAD generation."

The word "best" here is improper. About compression ability among different representaion, In Figure 4, it seems that quartiles and of the tokenized CAD is lower than CadQuery, which indicates that tokenized CAD is more compressed than CadQuery code.

The ablation in Section 7.1 and Table 3 only changes the choice of base model. Even though the authors claim that "Pre-training on code yields a clear performance gain over random initialization, demonstrating that the improvement stems from the prior knowledge
encoded in the model", the representation used in this part is all python code. There is no ablation on CAD generation performance over different CAD representations. The paper says that "CAD is best represented as code" but it doesn’t prove or show why that’s true.

2. Represent CAD as code is not rare, which may harm the novelty and contribution of this work elaborated in Lines 63-65. Such as following papers, which should all be included in related work:

- Cad-recode: Reverse engineering cad code from point clouds, Rukhovich et al. (https://arxiv.org/abs/2412.14042)

- Generating CAD Code with Vision-Language Models for 3D Designs, Alrashedy et al. (https://arxiv.org/abs/2410.05340)

- CAD-Assistant: Tool-Augmented VLLMs as Generic CAD Task Solvers, Mallis et al. (https://arxiv.org/pdf/2412.13810)

- From Idea to CAD: A Language Model-Driven Multi-Agent System for Collaborative Design, Ocker et al. (https://arxiv.org/abs/2503.04417)

- CAD-Coder: An Open-Source Vision-Language Model for Computer-Aided Design Code Generation, Doris et al. (https://arxiv.org/abs/2505.14646)

- CAD-Coder: Text-to-CAD Generation with Chain-of-Thought and Geometric Reward, Guan et al. (https://arxiv.org/abs/2505.19713)

- CAD-Coder:Text-Guided CAD Files Code Generation, He et al. (https://arxiv.org/abs/2505.08686)

3. Metrics such as CodeBLEU, NC, and IoU are missing for many baseline methods in Table 2, making the reported values difficult to interpret without comparable results across all methods.

**Minor problems:**

1. Missing comma in Line 51:

> "Unlike prior works that restrict evaluation to modalities already well supported by off-the-shelf LLMs our method uses prefix tuning (Li & Liang, 2021) to easily adapt our model to new input modalities instead of relying on fine-tuning a pre- aligned model."

2. Missing full stop in Line 212:

> "tokens compatible with DeepSeek-Coder-1B We employ a contrastive alignment loss of the form:"

3. Equation notation typo:

Lines 217-218:

> "and $sim$ is the cosine similarity function."

$sim$ is not align with "sim" in Lines 213-215:

$\mathcal{L}_{\text{align}} = - \log \frac{\exp(\text{sim}(z_g, z_c)/\tau)}{\sum_j \exp(\text{sim}(z_g, z_j)/\tau)},$

4. The description about CAD-MLLM seems to be wrong among whole passage. As far as I know, it's tokenized CAD-based method, which is similar to command sequence proposed in DeepCAD. And CADmium is json-based as far as I know, which is not discussed:

- DeepCAD: A Deep Generative Network for Computer-Aided Design Models, Wu et al. (https://arxiv.org/abs/2105.09492)

- CAD-MLLM: Unifying Multimodality-Conditioned CAD Generation With MLLM, Xu et al. (https://arxiv.org/abs/2411.04954)

- CADmium: Fine-Tuning Code Language Models for Text-Driven Sequential CAD Design, Govindarajan et al. (https://arxiv.org/pdf/2507.09792)

These parts need to be corrected, Line 40:

> "Recent work represents CAD as JSON and adapts general-purpose LLMs (Wang et al., 2025; Xu et al., 2025)"

Table 1, Lines 115-116:

> "Fine-Tune JSON Xu et al. (2025); Wang et al. (2025)"

Appendix, Lines 723-724:

> "(1) JSON-based representations as in Wang et al. (2025); Xu et al. (2025),"

5. Speed of StableLM-3B as base model is not reported in Table 3.

**Questions:**

1. In Lines 25-26:

> "We also contribute a dataset of 180k CAD code samples derived from Omni-CAD."

The reviewer is a little bit confused about the description here. Is "Omni-CAD"(180k) refers to dataset in CAD-MLLM(they claim to have 450k models) or from commercial company(https://www.omnicad.com/)?

If they are from commercial company, will they get released? If they are from dataset in CAD-MLLM, what is the exact contribution mentioned here in Abstract?

---

### Official Review · Reviewer_5ERW · 2025-11-01

**Soundness:** 1
**Presentation:** 3
**Contribution:** 3
**Rating:** 2
**Confidence:** 4

**Summary:**

This paper introduces Tiny-CAD-Coder, a lightweight multimodal CAD generation framework designed for efficiency and extensibility. Instead of building upon a large general-purpose LLM, the authors start from a code-oriented pretrained model and apply prefix-tuning with an aligned adapter to achieve cross-modality fusion among text, images, and parametric CAD sequences. The system aims to reduce model size and training cost while maintaining competitive performance across CAD-related tasks.

**Strengths:**

The unified coding framework could serve as a compact CAD foundation model, compatible with downstream finetuning for text-to-CAD, image-to-CAD, or sketch completion tasks. It shows great research potential.

The decision to begin from a code model (rather than a language model) is well-justified. CAD parametric sequences are structurally closer to programming syntax than to free-form natural language.

The model’s compactness is an appealing strength. Tiny-CAD-Coder demonstrates that CAD multimodality can be achieved without scaling to billions of parameters. The aligned adapter and prefix-tuning design simplify fusion without heavy architectural coupling. This approach is well-motivated for industrial adoption where compute and latency are concerns.

**Weaknesses:**

This work overlooks a significant number of relevant studies across various CAD generation tasks. In sequence completion, SkexGen [1] and HNC-CAD [2] should be discussed, while FlexCAD [3] is particularly noteworthy for introducing LLMs into the CAD domain and supporting completion tasks. In text-to-CAD generation, the omission of CAD-Llama [4] and CADFusion [5] is critical, as both represent strong and competitive baselines. Although I am less familiar with image-to-CAD and BREP-to-CAD tasks, works such as Vitruvion and CAD-Diffuser clearly align with similar directions. It is also worth noting that not all of these approaches rely on LLMs, suggesting that Tiny-CAD-Coder may not necessarily be the most lightweight solution among them.

The experimental section is also far from sufficient. First, several key baselines are missing. Second, many quantitative results are absent from Table 2. While one could argue that some metrics were not included in the original papers, several metrics such as IoU are clearly feasible to compute. As a result, the current evaluation does not provide a comprehensive understanding of the proposed framework or its advantages over prior work; indeed, the presented results indicate mostly marginal improvements. Third, the qualitative evaluation is limited: text-to-CAD results are missing, and none of the qualitative figures include baseline comparisons.

[1] SkexGen: Autoregressive Generation of CAD Construction Sequences with Disentangled Codebooks. ICML 2022.

[2] Hierarchical Neural Coding for Controllable CAD Model Generation. ICML 2023.

[3] FlexCAD: Unified and Versatile Controllable CAD Generation with Fine-tuned Large Language Models. ICLR 2025.

[4] CAD-Llama: Leveraging Large Language Models for Computer-Aided Design Parametric 3D Model Generation. CVPR 2025.

[5] Text-to-CAD Generation Through Infusing Visual Feedback in Large Language Models. ICML 2025.

[6] Vitruvion: A Generative Model of Parametric CAD Sketches. ICLR 2022.

[7] Draw Step by Step: Reconstructing CAD Construction Sequences from Point Clouds via Multimodal Diffusion. CVPR 2024

**Questions:**

The authors should more thoroughly discuss the connections between their work and existing research, addressing recent progress in related areas and providing necessary comparisons with prior methods.

Additionally, more qualitative results are needed. These should include examples from the text-to-CAD domain as well as visual comparisons with baseline methods. It is also important to incorporate additional baselines, as mentioned previously, to better contextualize the model’s performance.

*While my concerns regarding the empirical comparisons are serious, I appreciate the insights and technical contributions presented in this work. I would be glad to reconsider my evaluation and raise my score if the authors strengthen these sections in a revised version.

---

### Official Review · Reviewer_f3X5 · 2025-11-03

**Soundness:** 3
**Presentation:** 3
**Contribution:** 1
**Rating:** 2
**Confidence:** 5

**Summary:**

Authors proposed to use pretrained code-llm for generating CAD construction history. They showed that rewriting CAD operations in python-like format is more suitable for LoRA finetuning deepseek-coder-1b. They also proposed to use contrastive alignement and prefix tuning to support multimodal control.

**Strengths:**

Reformulating DSL into python-like code when the base model is pretrained LLM sounds like a promising solution.  The alignment and prefix finetuning are also reasonable solution to support multimodal understanding. Overall the pipeline is clean and sound. Results are also SOTA compared to other models in this line of works.

**Weaknesses:**

Using more code-like or python-like data format for CAD and paired it with LLM is not new. Many recent works have explored this under the CADQuery python code format. This paper lacks a detailed explanation of why their format is better than CADQuery, so it is not immediately clear to me where the improvement comes from. From the ablation it seems that the base model matters a lot, this is worrying because all these models use different base model or even train it from scartch, so it is hard to fairly compare.

The omni cad dataset is fairly simple. It would be more convincing if authors tried it on WHUCAD or other more complex dataset. Also some evaluation is not well justified, e.g. invalid ratio. When it comes to CAD construction format, as long as the sketch is closed, pretty much everything will be somewhat a valid shape. The real bottleneck with this approach compared to direct B-Rep generation is the complexity. Simple cuboid / sphere-like shape, or those from one step of sketch-and-extrude is not interesting and far from real-world scenerio. Usually when it comes to sequence-based, I favor those that can generate more complex shapes over valid ratio. The results provided in this paper however is very simple and not impressive.

Some details are missing in the paper. 1) how is the code and geometric embedding encoded? 2) How is the gradient backprop from execute in the uv-net reconstruction loss?

**Questions:**

How is the output complexity and diversity compared to previous methods?

How is the proposed python-format better than CADQuery?

Some implementation details are missing and should be added.

---

### Note · Program_Chairs · 2026-01-17
**Submission Desk Rejected by Program Chairs**

The following references in this submission do not refer to real documents and/or have major errors in bibliographic information:

 Elias Berger, Felix Braun, Jan Mehlstäubl, and Kristin Paetzold-Byhain. Task-adaptive cad generation via decoder-only pretrained transformer. arXiv preprint arXiv:2501.00000, 2025a.